# Religious Singing in Kashubia: Tradition and Modernity

**Jan Perszon**

Faculty of Theology, Nicolaus Copernicus University, Gagarina 37, 87100 Toruń, Poland; jan.perszon@umk.pl

**Abstract:** The article answers the question of how the custom of religious singing was created in Kashubia and constitutes an integral part of the worship of the Church and local piety/religiosity until modern times. Nowadays, musical worship is manifested both in family and rural life (religious singing at home, May devotions, Rosary prayer services, vigil for the dead) as well as in ecclesial sphere (church services, Calvary celebrations, feasts, processions, and pilgrimages). It integrates local communities and simultaneously "broadens" their relationship with the sacred (God). It is important to present the process of selecting leaders of folk singing, as there is no institutional support in this matter such as music schools, organ courses or choirs. It can be assumed that, as was the case in the nineteenth and twentieth centuries, the family tradition plays a decisive role in leading and animating collective singing. The changes which are taking place in this respect now require further research and may be the result of the transformations in the field of folk religious singing in Kashubia, which are the consequence of the dynamic social transformation in this region. This study is based on ethnographic field research conducted by the author in 2016–2021 and the literature on the subject.

**Keywords:** Kashubia; religious singing; folk piety/religiosity; performance

## 1. Introduction

Kashubians, inhabiting the northern part of the Pomeranian Voivodeship, are the largest Polish ethnic minority with around 500,000 citizens. As descendants of Slavic Pomeranians, they spoke their own language. Today, the Kashubian language has the status of a regional language and has its own rich literature (Synak 1998; Kleina and Obracht-Prondzyński 2012). Until the end of the 20th century, almost exclusively Polish was used in prayer and song in Kashubia. As research has shown, the oldest relics of religious songs in Kashubian date back to the 16th and 17th centuries (Sikora 2006; Frankowska 2015). In this article, we do not deal with Kashubian carols and pastorals, excellently arranged by Witosława Frankowska (Frankowska 2015), nor with Kashubian sung folklore (Bielawski and Mioduchowska 1997–1998). The author of the article raises the question of how the formation of the popular, Catholic singing tradition in Kashubia took place. How did it function over the past few hundred years, what model of religiosity did it form, what contributed to its continuation by subsequent generations, and, finally, what has contributed to its rapid decline? The answer to these questions will be based on the Polish (and, thus, Kashubian) ethnomusicological, ethnographic, liturgical and theological literature. An important source is the ethnographic field research conducted by the author between the years 1992–2022 (interviews and participant observation). It should be added that, as far as the research of the so-called popular piety (the term is highly imprecise; see Mazurek and Potulski 2020) on Kashubia is concerned, it is, so far, unique.

Folk religious singing in Kashubia has a long history and over the last few centuries, it had an important culture-making function. For entire generations, it became a school of religious life, a specific type of catechism (Christian initiation) and a medium of Polish cultural and national identity during the Prussian occupation. Until recently, the area in which folk singing could be treated as public performance was in Church, in family and neighbourly life, that is, devotions or parochial and regional occasions celebrated in a liturgical year (for example, processions, pilgrimages, devotions in Kalwaria Wejherowska,

May and June devotions by roadside shrines). The "folk religious song" covers ecclesiastical (liturgical) and non-liturgical songs as well as non-ecclesiastical songs which are not connected with a liturgy but form a part of collective and private acts of devotion (Bartkowski 1987, pp. 24–25; Zoła 2010, pp. 369–70). Therefore, we do not refer here to "authentic" folklore which is associated with the artist's anonymity as well as the orality and variability of the message (Zoła 1992, pp. 86–87). Non-liturgical religious songs form the largest part of folk sacred repertoire conveyed in living tradition. For the purpose of categorising them, there are three cycles featured in Polish ethnomusicology: the liturgical year cycle, the weekly cycle and the daily cycle (Zoła 2010, p. 371)[1]. Their analysis and presenting the chants performed within them (as well as the defined function of amateur singers) go beyond the narrow range of the present study. Therefore, we present only some of them. The liturgies and Church ceremonies are extended with the so-called folk piety, namely the customs and devotions practised within the family (home liturgy) or in a wider neighbourly or rural community (Bartkowski 1987, pp. 52–57).

Until the present times, an important role in creating and performing folk religious singing has been fulfilled by leaders. Their love for singing and aptitude for leading common rituals and devotions has almost always been developed in the family environment. In many cases, the singing tradition has lasted within families for a few generations. In Kashubia the "profession" of a singer has virtually always been practised by men; hence, the prayerful tradition is passed from father to son. We present two examples of such still actively "singing" families.

At the end of the article, we make a short attempt to show the social and religious transformation which results in the decline of old singing culture and its social functions. Changes occurring in a social system, especially over the last fifty years, exert a significant influence on the continuation (or stagnation and abandonment) of traditional piety. Therefore, we discuss some of the aspects which influence the transformation of folk singing tradition in Kashubia and the attempts to introduce it into the world of artistic events.

## 2. The Formation of Folk Devotional Singing in Poland and Kashubia

Folk religious (Christian) singing appeared in Kashubia as a result of Christianising Pomerania in the 10th/11th centuries. Preserved historical sources indicate that the first religious singing in the indigenous language (the literary Polish language did not exist then) was performed in the form of "a sung prayer", probably including the Lord's Prayer, 10 Commandments and the Apostles' Creed (Bielawska 1980, p. 122; Korolko 1980, pp. XI–XV; Michałowska 1995, pp. 270–72; Wydra 1973, 3 n.). In Pomerania, the prayer was chanted, just as in churches in other regions of Poland, during Holy Mass (before or after the sermon), until as late as the 17th century (Hinz 1994, p. 40)[2]. For several hundred years, liturgical (Gregorian) singing in Poland was performed only in Latin. A few manuscripts indicate that at princely courts, prayers were chanted (and sometimes even created) in Polish as early as in the 11th and 12th century; however, it happened only among the elite of the contemporary society (Korolko 1980, pp. XIV–XV)[3]. The creation of Slavic chants and their perfomance in Church life was strongly influenced by a constant flow of Germanic people into Polish lands, which, according to the leaders of the Polish community, posed a threat to people's identity. Polish songs could only be cultivated in parochial or collegiate church schools. Therefore, the Archbishop of Gniezno named Pełka (between 1233 and 1246) imposed a ban on designating Germans as teachers, and the Synod of Włocławek of 1248 ordered the priests to say the Lord's Prayer and confession of faith in Polish each Sunday after the sermon (Korolko 1980, pp. XIII–XIV; Bielawska 1980, p. 122)[4]. The Hail Mary prayer (the Annunciation) was already known among Polish people (in its full version, with the names of Jesus and Mary) in the 13th century (Kopeć 1997, pp. 331–32)[5]. Hieronim Feicht (1975, p. 82) supposes that in the 13th century the recitation turned into singing.

A little bit later (13th–14th century), the first sparse chants in the Polish language started to be composed (one of them was *Bogurodzica*—"the Mother of God"). At first

they were translations of Latin hymns and sequences (Korolko 1980, pp. XII–XV)[6]. "The Mother of God", similarly to other chants (refrains), for example, those of Easter, is a conglomerate of traces. Some of them instantly became popular and are sung to this day, for instance, *Chrystus zmartwychwstał jest* (Eng. "Christ Is Risen"), *Przez twoje święte zmartwychpowstanie* (Eng. "Thanks to Your Sacred Resurrection"), *Krzyżu święty* (Eng. "Holy Cross") (Pawlak 2000, pp. 146–49; Feicht 1975, p. 310; Feicht 1958; Korolko 1980, pp. XXX–XXXVII). A similar process concerns sequences. In medieval Poland, plenty of them were created in Latin. However, shortly after, they were translated into Polish so that they could become folk songs. A large number of textual and musical variants indicates that their process of assuming the folk character started very early. It proves that in spite of the Church control over the liturgical repertoire, local performing traditions were tolerated for a long time (Zoła 2010, p. 370). Korolko (XV–XIX) emphasises the importance of social transitions (for example, in the form of the inception of mendicant orders: the Dominicans, the Franciscans, the Carmelites and the Augustinians) as well as theological and spiritual changes (discovering the humanity of Jesus, his birth and Passion, recognising mysticism and love as a way of gaining an insight into God). They enabled the widespread evangelisation of Polish people. The new orders were exactly the ones which indirectly became the promotors of new spirituality and new national religious songs.

Religious singing flourished in the 16th–17th centuries when the lyrics were written, and the melodies were composed mostly in monasteries (Franciscans, Bernardines, Jesuits). As early as the 15th century, the Bernardines published numerous devotional books and meditations in Polish (Brückner 1904, 118 n.; Michałowska 1995, pp. 341–82). An outstanding role in the creation of Polish songs was played by a Bernardine monk, blessed Ladislaus of Giulio (died 1505), who wrote a series of religious poems and songs concerning the Passion of Christ and Mother of God. As a provincial superior, he issued an order to sing them in Poland, which is why they spread so quickly (Korolko 1980, pp. XIX–XX)[7]. In the late Middle Ages, the activity of laymen increased highly, mainly thanks to Church brotherhoods and tertiaries. These communities contributed to the publication of *Kancjonał kórnicki* (Eng. "The Hymnal of Kórnik") and *Kancjonał Puławski* (Eng. "The Hymnal of Puławy") (Korolko 1980, pp. XXI–XXII; Bielawska 1980, pp. 123–25). Folk songs (not only Christian in their content) were sung in some periods of a liturgical year (Christmas and Great Lent during theatre stagings, which the society of the late Middle Ages delighted in (Korolko 1980, pp. XXIV–XXIX)[8]. During the Renaissance, liturgical and ecclesiastical Polish songs were already widespread, which is confirmed by hymnals. The Baroque singing works were extremely abundant, frequently not theologically correct but easy to learn. Therefore, after the Council of Trent (1545–1563), Polish bishops started to issue decrees restricting the inflow of new and unsanctioned songs (Bielawska 1980, pp. 129–31). In Poland, strict liturgical restrictions, imposed on local churches by the above mentioned Council, were accepted and introduced only partially (metropolitan synod of 1577 in Piotrków). One of the accommodations was the universal approval of folk singing during liturgies. Liturgical books published after the Council of Trent (Psalterium, Graduale, Antiphonarium, Processional and the so-called Ritual of Piotrków of 1628) allowed Polish songs to be sung, often performed alternately with the Latin ones (Wit 1980, pp. 218–25). Subsequent diocesan synods recommended performing Latin chants and forming chanting groups (Wit 1980, pp. 225–28). In the 19th century, classical liturgical (chorale and Gregorian) singing fell into decay almost in all dioceses of the former Polish Republic. In turn, Polish pious songs developed, which is confirmed in the hymnals published since the middle of the 19th century (Wit 1980, pp. 233–54)[9].

A simple melody and rhythm of consecutive verses made them easy to remember. This was also the time when, due to the Catholic reforms (the Counter-Reformation) struggling with Protestantism, religious singing was popularised in society and the quality of religious music increased (Perz 1992, p. 33). In response to the Protestant threat, collections of catechism songs were published, which were supposed to deepen people's devotion (Bielawska 1980, pp. 127–28)[10]. A special role was played by Jesuits, for whom Catholic

songs were a tool used for teaching and a remedy for dissidents' canticles (Nowicka-Jeżowa 1992, pp. 302–9). The 17th century witnessed the appearance of the practice of singing Sundays and festive vespers in Polish. They were memorised and became an element of orally transmitted tradition (Bartkowski 1987, pp. 118–20; Zoła 2010, p. 371). In the 17th century, bishops in plenty of dioceses issued an order to sing the Rosary, Hours of the Virgin, the Litany of Loreto and the Litany of the Saints before Sunday Mass, which displaced the Liturgy of the Hours in some places, that is, vespers (Wit 1980, pp. 228–31)[11]. When it comes to non-liturgical songs, we must mention the Hours of the Passion, the Little Office of the Immaculate Conception and Lamentations (Korolko 1980, pp. XLII–XLIV; Zoła 2010, p. 371; Wit 1980, pp. 263–64)[12]. Little hours practised as officium parvum were prayed in monasteries until the 16th century, when they were printed in the form of a Polish-language book and became a Saturday Catholic devotion (Kopeć 1997, pp. 323–25)[13]. We must remember that since the middle of the 16th century, the entire Catholic piety in Poland developed in confrontation with Protestantism, which certainly had an influence on the evolution of some devotions and customs, for instance, the cult of Mother of God or the pilgrimage movement (Kopeć 1997, pp. 416–418; Roszak 2020). An additional element of Polish Catholics' spirituality was the awareness of being the Bulwark of Christendom (Catholicism), *antemurale christianitatis* (Tazbir 1984; Borkowska 1982). In the 16th century, Poland struggled with the Orthodox Church in the East, Islam in the South and Protestant Prussians in the North and West. Such a geopolitical situation favoured the identification of the Catholic tradition and ancestors' faith as a defence against strangers. The belief of Mary's protection for the "Catholic people" was enhanced in the face of numerous wars and threats leading to the acknowledgement of the Mother of God as the Queen of Poland (Kopeć 1997, pp. 419–27). A tension between Catholics and Protestants could also be observed in Kashubia (Klemp 1993).

In the 19th century, Lamentations became the favourite devotion. They took the place of polyphonic Latin passions which had been known since the 15th century[14]. The oldest Polish carols appeared together with Nativity plays as early as the 14th century (Korolko 1980, pp. L–LIV; Perszon 2019, pp. 212–14). In turn, Catholic mournful funeral psalms, which were already created in the late Middle Ages (as translations of Latin hymns), started to be commonly applicable (both by the people and the Church) in the middle of the 17th century (Nowicka-Jeżowa 1992, pp. 297–309)[15].

Marian songs, especially, gained great popularity, which was correlated with the foundation of local sanctuaries dedicated to the Mother of God (Kopeć 1997, pp. 313–316). It is worth mentioning that in 17th- and 18th-century Poland, there were several hundred local sanctuaries and a few whose range was more extended (Częstochowa, Święty Krzyż—Passion sanctuary, Kalwaria Zebrzydowska, Saint Anne's Mountain in Silesia) (Witkowska 1982, 1986, 1995; Mróz 2021, p. 1; Kopeć 1997, pp. 391–405). Many regions of Poland witnessed the popularisation of Marian chaplets dedicated to her joys and sorrows (Kopeć 1997, pp. 336–53)[16]. However, in the 20th century, they were not too popular in Kashubia. Just as in the entirety of Poland, an honourable place in non-liturgical devotion was held (probably since the 16th century) by the Rosary. The Rosary was developed around 1409 by two Carthusians from Trier (Adolf from Essen and Dominic from Prussia) as a meditative prayer (*Rosarium Beatae Mariae Virginis*) and instantly penetrated into the Republic of Poland (Kopeć 1997, pp. 353–62)[17]. The Rosary brotherhoods (confraternities), which were founded then, are perhaps the only devotional associations which have continually existed in the mass form until these days. The habit of carrying a rosary by members of brotherhoods and monks and putting it inside coffins dates back to the 15th and 16th century (Kopeć 1997, p. 362).

In Vistula Pomerania, which belonged to the Pomerania Archidiaconate of the Diocese of Włocławek with the exception of the Diocese of Chełmno, in the 16th–17th centuries, there were no places of renown of the Marian cult (except for in Zamarte and Pruszcz Pomorski villages) (Witkowska 1986, p. 519). In the 18th and 19th centuries, the status of a sanctuary with more than just regional importance was acquired by a Marian place

in Łąki Bratiańskie, bringing it more than just regional importance (Korecki 2002; Piszcz 1971). Since the middle of the 17th century, more importance was attached to Kalwaria Wejherowska with its Passion cult (Więckowiak 2006; Więckowiak 1982; Kustusz 1991, 76–106; Walkusz 2014), and, to a lesser degree, to the Marian village of Swarzewo since the 16th century (Pryczkowski 2019; Szulist 1974, p. 55). In the 20th century, the south of Kashubia witnessed the foundation of Kalwaria Wielewska, which has been a local centre of the Passion cult until the modern day (Borzyszkowski 1986, pp. 171–248; Perszon 2016, pp. 57–60). For several hundreds of years, these sancturacies were a "school" of religious singing for Kashubians in their native language.

Preserved sources allow us to conclude that the Catholics from Pomerania and Kashubia contributed, especially after the Peace of Toruń (1466), to the development of Polish-language songs in the same way as the inhabitants of other regions. The Synod of Riga of 1428 (the Diocese of Chełmno belonged to the Province of Riga) required priests to speak the native language of their congregations and not to be "mute shepherds" (*pastores muti*) (Hinz 1994, p. 55). The "Mother of God" hymn was known and chanted (as recommended by the bishops of Chełmno in the 18th century). Pomeranian and Warmian sources from the late 16th century report the widespread use of Polish songs during services. The Synod of Gdańsk of 1585 allowed chanting of folk songs on the condition they were old and approved by the Church, and the oldest preserved hymnal in Chełmno (dating from 1660) includes almost exclusively Polish Mass songs (Hinz 1994, pp. 40–44). In Chełmno, the common singing of Polish songs (*pias canticulas*) was introduced at the end of the 17th century. People sang Polish vespers and numerous little hours and offices during each procession (Easter, Corpus Christi, Way of the Cross) and also during masses. For example, during the Chaplet of the Holy Trinity, the Holy Sacrament, the Passion chaplets, on Sundays and holidays, Hours of the Virgin, Hours of Saint Roch and Saint Vincent de Paul, and for the first time, Lamentations in 1724 (Hinz 1994, pp. 75–77). The richness and diversity of Polish songs in Kashubia is definitely proved by the hymnal of *Pious Songs*, the manuscript of which was written in Wejherowo between 1711 and 1732. It has two volumes and includes as many as 600 chants (2/3 of them are Polish). Apart from periodical songs, it also includes Marian chants, songs in honour of the Saints, morning, afternoon and evening songs, penitential and funeral psalms, as well as other chants for various occasions (Hinz 1994, p. 44)[18].

In the 19th century, May devotions started to be prayed in Poland. They were initiated by Jesuits in Tarnopol (1826), and by the middle of the century, they had already become popular in a number of dioceses. In the Diocese of Włocławek, it was introduced in 1859, in the Diocese of Sandomierz in 1860, and in the Archdiocese of Warsaw in 1863. The devotion involved singing the Litany of the Blessed Virgin Mary and other Marian chants (Wit 1980, p. 265). Simultaneously, Mass songs, performed by an organist and congregation, were developing dynamically. A vast number of "Polish services" included in the hymnals from the second half of the 19th century indicate that local traditions were of considerable importance (Wit 1980, pp. 267–69). At that time, numerous editions of songbooks were published in the Diocese of Chełmno (since 1821, it also covered the Pomeranian Archidiaconate). The most significant ones were *Zbiór pieśni nabożnych katolickich do użytku kościelnego i domowego* (Eng. "Collection of Pious Songs for Church and Home Purposes"), devised by Father Szczepan Keller (Keller 1871), and *Melodye do zbioru pieśni nabożnych katolickich dla użytku kościelnego ułożone do grania na organach i śpiewania na cztery głosy* (Eng. "Melodies to the Collection of Catholic Pious Songs for Church Purposes Composed for Organs and 4-Part Choirs"), devised by Father Józef Mazurowski and Teodor Kiewicz, intended for organists and chorus singers (Mazurowski 1870)[19]. Therefore, at the end of the 18th century, the singing movements were highly influenced by subsequent editions of songbooks and prayer books. It is symptomatic that even though the actual elimination of illiteracy happened as late as the end of the 19th century (Klotzke 1980, pp. 65–104), very popular and commonly used hymnals had been printed (in Polish) much earlier, for example, the calvary prayer books *Droga do* nieba (Eng. "Road to Heaven"),

which had been published by the Observant monastery in Wejherowo in 1717 (Kustusz 1976)[20]. One hundred years later, weighty and arguably expensive hymnals started to be published also in Pelplin. Their consecutive editions prove that there was a high demand for them[21]. A separate "genre" of folk singing books were funeral songbooks[22]. They were smaller and convenient. Polish hymnals also became one of the factors establishing the cultural (and, consequently, national) identity of the Silesian people (Reginek 2012, pp. 3–16; Górecki 1994, pp. 92–96).

It is no exaggeration to say that the 100-year period between the second half of the 19th century and the middle of the 20th century is the peak of the "pious" singing culture in Kashubia. Religious singing was an integral part of personal, familial and social life. Such a state of things resulted not only from the organic work of the clergy but a high level of religiousness in the whole of society, which also included the Protestants who constituted about 20% of the population (Labuda 1995, pp. 107–9; 141–42). The anti-Catholic and anti-Polish character of the Prussian authority (lately German in 1871) could certainly clash with Kashubians and Pomeranians. This was due to the fact that they were linked with Catholicism and the Polish language with the culture eradicated by the occupants of the Republic of Poland (Mazurek and Potulski 2020, pp. 30–33; Perszon 2020). Similar processes also took place in Upper Silesia, where Prussian authorities harassed pilgrimage movements (Górecki 1994, pp. 96–103). Nevertheless, one can suppose that a significant factor creating the foundation for national awakening and awareness of being different from the German element was the economic reform conducted by the Prussian government between 1807 and 1850. Gerard Labuda (1995, pp. 138–41) emphasises the crucial meaning of the abolition of serfdom in Kashubia. It set foundations for the formation of the free peasantry class and, much later, civil society. Free farmers, together with their families, became not only the owners of their lands, buildings and animals but also independent, often rich, people who could decide on their owns fate and were aware of their dignity and rights. This economic and social status, combined with Christian virtues (diligence, honesty, thriftiness, piousness) in co-operation with the clergy, enabled full religious development. It seems that this social structure accounts both for the Pole-Catholic cluster as well as the diversity of family and rural religious rituals (Mazurek and Potulski 2020, pp. 32–33)[23]. At the end of the 18th and the beginning of the 19th century, Polish social elites succumbed to liberal slogans, whereas in the second half of the 19th century, the positivistic "basic work" started to become popular. While it was initiated by Polish (often clergy) intelligentsia, it originated mostly in the countryside (for example, in Greater Poland or Pomerania) (Górski 1986, pp. 302–12). It would not have been possible without the economic and spiritual subjectivity of rural elites, that is, of the farmers.

The learning of songs by people was based on memorisation, as only a few elites were literate. The ability of singing from memory was impressive. At the turn of the 20th and 21st century, it was still possible to meet elderly people (both men and women) who were able to sing several hundred multi-verse songs from memory (Mills 2005)[24]. Thus, the melodies and verses memorised in childhood were "owned" by the singer's soul. Their contents shaped religious imagination, formed individual and collective piousness, existential attitudes, popularised hagiographies and expressed emotions (Perszon 2017, p. 189). For centuries, religious songs have been one of the principal components of Catholic and popular spirituality (Nowicka-Jeżowa 1992, 302 n.; Kolbuszewski 1986).

### 3. Family as School of Religious Singing

The dissemination of religious songs in Kashubia is associated with several interacting factors. The first is the organic work of the Roman Catholic Church, which, as shown above, systematically and persistently evangelized the Slavic population through singing. The main difficulty that Catholicism had to face was its attachment to the Latin liturgy. In Poland, similarly to other countries, the tension between the Latin liturgy and folk religious singing (as well as non-liturgical services) lasted until the reception of the Second Vatical Council. As we have seen, the literary Polish language stabilized only in the 15th

century, enabling the development of native religious songs. Another stimulus, already in the 16th and 17th centuries, was the Reformation, which used the native language of the faithful in its worship. In a way, it "forced" (despite the strict rules of the Latin-promoting Council of Trent) the acceptance of folk singing in the local Church, as well as the dissemination of printed "aids" in the form of prayer books and hymnals. Religious singing (initially in Latin) was promoted through church education. For centuries, the Church obliged parish administrators to run schools (usually elementary ones), where teachers were often organists (Wojtkowski 1969, pp. 38–39; Hinz 1994, p. 49; Klotzke 2017, pp. 29–32). A higher level was represented by monastic schools run by monasteries (Kustusz 2006, pp. 77–131)[25]. Teachers devoted a significant part of their lessons to teaching singing[26]. As mentioned before, already in the late Middle Ages, during Sunday Mass, people without the knowledge of Latin said common prayers and sang popular songs in their local language. The basis for singing together was memorisation, remembering texts and melody (Bartkowski 1992, p. 81; Szulc 2007, pp. 114–19). The Catholic Church preferred polyphonic liturgical chants. Although after the Council of Trent, especially Gregorian ones (in Latin), these were performed by bands or qualified cantors, but only in major municipal churches. People (both from villages and towns) sang in Polish. The divergence of popular and official piety (ecclesiastical vs. Latin) resulted in the development of religious songs and an increase in the number of pious customs (rituals) and services functioning on the margins of the liturgy.

One can assume that pieces of music learnt in the parish church were later performed at home and during work in the field. At the end of the 19th century in Pomerania, as in the whole of Catholic Europe, the Cecilian movement played an important role in singing culture (Wit 1980, pp. 274–84; Hinz 1994, pp. 223–29; 262–82)[27]. In many parishes, choirs were established that gathered and formed the singing elite of the local community (Klotzke 2009, pp. 42–45; Pryczkowski 2016, p. 145). In the Diocese of Chełmno, the Cecilian movement (in existence since 1869) developed dynamically thanks to the enthusiasts of Church music working in Pelplin (Fr. Józef Mazurowski, Fr. Bernard Ruchniewicz, Fr. Wacław Lewandowski). In 1887, an organ school began its operation in Pelplin. This significantly raised the level of Church music in the parishes (Hinz 1994, p. 252).

These institutional measures transferred the art of praising God through common singing to families and neighbourhood communities. Some Kashubian families can boast about an over-hundred-year-old singing tradition (because the memory does not go further than that). These are farming families. *Stabilitas loci*, local parish, singing traditions, wise activities of priests and organists in the field of popularising musical culture and living piety, created an atmosphere in which "praising God with singing", as the informants put it, was something convenient and natural.

Maintaining their own households and cultivating their own fields encouraged Kashubian farmers to have many children. Almost all informants covered by the research in the 1990s and in 2015 (Perszon 2017; Perszon 2015b) came from large families.

The custom of singing in families was of great significance in the process of the dissemination of religious songs. Even in the first half of the 20th century (before the electrification of villages and the dissemination of electronic media–radio), in most Kashubian houses, especially in the months when field work ceased (September–March), family religious songs were sung every evening. The Klein family, residing in Barłomino (Sampowskie) since the end of the 18th century (Grzegorz Klein established the beginnings of the Kashubian branch of the family, having originated in East Prussia) and throughout the 19th and 20th centuries, was famous for its male singers[28]. The tradition of religious (but also patriotic and Polish) singing was passed down in these families from generation to generation. In modern times (the 20th century), two members of the family, Bernard and Herman Klein, as distant cousins, performed the important function of leaders of Church and folk (family and neighbourly) services. Both of them conducted the Hours of the Virgin Mary in the church before Mass every Sunday. Bernard (1881–1951) led (until the 1950s) the Stations of the Cross service in the church in Luzino; Herman (1888–1981) led all rural services, especially

funeral services (empty nights). Additionally, having been blessed with a powerful voice, he celebrated the first funeral station in the mourning house and led the funeral procession to the parish church. Their siblings, children and grandchildren, to a large extent, joined the local Lutnia Choir (before and after World War II), and nowadays, they actively participate in the animation of the "empty nights"[29]. According to the informants, with the exception of spring and summer, religious songs were sung together in both families every evening (while sitting by the kerosene lamp). In October, the Rosary and Marian songs; in November, mourning songs (for the people who had passed away); in December, advent songs and then carols and pastorals. During Lent, Passion canticles were sung every day, and Lamentations were sung on Sunday afternoons. Parents and grandparents, after getting up, sang "Hours of the Virgin Mary"[30] every day from memory. The last surviving daughter of Herman Klein says that on Resurrection Sunday, her father would wake the family up for Holy Mass by going through all the rooms and singing with a powerful voice *Wesoły nam dzień dziś nastał* (Eng. "A Joyous Day Has Come"). When participating in some family celebrations (wedding, baptism, First Communion, funeral), he also used to sing the hymn *Te Deum* (Eng. "Thee, Oh God, We Praise")[31].

The Pryczkowski family, who still lives in Stążki in the parish of Sianowo (Kartuzy district), also nurtured the richness of religious singing. According to Eugeniusz Pryczkowski, singing traditions date back to at least the 19th century. The oral tradition of the oldest inhabitants of Staniszewo (which was probably the seat of the family) says that the Pryczkowskis were the village headmen for over two hundred years. The Frydrycan cadastre of 1773 states that the headman of the village, part of which belonged to him, was Stanisław Pryczkowski. In 1864, Jan Pryczkowski signed inventory documents establishing a parish in Sianowo, stating that he had at least five sons. One of his sons moved to Reskowo, giving rise to the local branch of the Pryczkowski family. Another settled in Zbychowo near Reda. The last of Jan's sons—Walenty (1840–1924)—settled on a farm on the outskirts of the hamlet of Stążki after marrying his wife Julianna (nee Zielonka). Yet another son inherited part of the Staniszewo estate, located on the so-called Kępie. Miotk (See: Pryczkowski 2012, pp. 6–7) married the widow of his son Józef. Three priests come from the Miotk family: Fr. Marian Miotk from Kępa and brothers Andrzej and Marek from Olszowe Błoto (currently the parish of Mirachowo) (Pryczkowski 2016, p. 251)[32]. Jan Pryczkowski (1882–1957), the grandson of Jan (headman of Staniszewo from 1864), was a member of the parish choir in Sianowo and would lead services and Rosaries for the deceased, processions to the church and the cemetery and May and June services at the local chapel. After him, the "office" of the singer–leader was taken over by his son Antoni. Nowadays, the tradition is carried on by his five sons. Jan inherited the farm in Stążki. Marek, Fr. Mateusz Pryczkowski's father, leads rosaries in Kolonia, a village where he has been living for years. In addition, sister Elżbieta Radtke leads Rosaries and songs for the deceased in the village of Mieroszyno and the surrounding area (Puck district). The other siblings also take part in such activities.

Antoni's brother, Stanisław Pryczkowski, led a Kashubian folk band in Staniszewo after World War II, and also organised theatre performances, e.g., art by Fr. Bernard Sychta, *Hanka sa żeni*.[33] Therefore, the Pryczkowski family is related to the Miotk family from Różny Dąb and Kępa, whose members were famous in the area for their participation in religious life at the Sianowo Church and village services such as May and June services, funerals and empty nights. Before 1939, members of both families were active in the Catholic Youth Association (Pl. Katolickie Stowarzyszenie Młodzieży (KSM)) in Sianowo (Pryczkowski 2012, p. 15). Weronika Ceynowa, nee Pryczkowska, daughter of Antoni's youngest son, Eugeniusz, is a graduate of the vocal faculty of the Music Academy and the author of several albums with Kashubian music. Janusz Pryczkowski, Jan's son, is an accordionist; during his years as a student, he played an important role in the orchestra of the academic band "Kortowo" in Olsztyn. He also willingly sings on empty nights, nurturing family traditions as a successor on the family farm in the parish of Sianowo[34].

The history of only two Kashubian "singers" families presented above shows that in the last century, a very important cultural function was played by settled farming families. The rhythm of their work and life perfectly harmonised with the liturgical calendar of the Catholic Church. In the family, there was not only social and cultural socialisation (with limited influence of school and parish) but, above all, a religious one. Its inseparable element was, in many cases every day, a common family evening prayer and singing Church songs. In the case when the distance to the church was considerable, there was a "home church", usually animated by the senior member of a multi-generational family. This does not exclude the existence of religiously and spiritually neglected families. At the end of the 19th century and in the first half of the 20th century, however, the "cultural fashion" in the form of extensive Catholic piety was imposed by zealous individuals and families, formed in the disciplined post-tridentine mentality.

**4. The Role of the Leaders of the Singing Movement**

Until the mid-twentieth century, and to a large extent also later, in almost every village or hamlet, there were lay Catholics (usually elderly people, men and women) who led local paraliturgical services. They also represented their local community and parish in religious events (e.g., pilgrimages) of a regional character. Usually, they were people with developed, deep personal piety (regular confession, frequent Holy Communion and participation in non-obligatory services), who were active in their parish and usually trusted by the local parish priest. Similar qualifications were possessed by "singers" (Pl. "śpiewocy") in the 19th and 20th centuries in Upper Silesia (**Górecki, p. 43**). Nobody designated them leaders of the collective prayers; they undertook these duties of their own accord, devoting a lot of time and effort to them. The "office" of a singer/leader of village services (although, until the 1990s, such people were also found in Kashubian towns such as Bytów, Kartuzy, Kościerzyna, Wejherowo, Reda, Rumia, Puck) was usually taken over by people from the middle generation following their parents or in-laws. The residents of the villages knew each other very well; hence, people undertaking religious duties enjoyed recognition and authority among them. The parish and the pious fraternities operating there were the school for such leaders. One could assume that the Rosary brotherhoods were of the greatest importance in this regard. It so happens that up until the present day, the Rosary in popular (folk) piety is prayer "for any occasion", the prayer *par excellence*[35].

Singers–leaders, until the 1980s, with the consent of local priests, performed important tasks in the life of the Church. Namely, they animated (conducted) services, which were celebrated periodically in the parish and the parish church. Throughout the liturgical year, they performed Hours of the Virgin Mary every Sunday (in many rural parishes, this tradition continues) before the commencement of the so-called High Mass. In many churches, women or men lead the daily rosary prayer (formerly only before the morning Mass) preceding the celebration of the Eucharist. In many places, the Litany of Loreto and a song adapted to the liturgical season were integral elements of this service. Half a century ago, in some churches, during Lent, lay people led meditations and singing of the Stations of the Cross celebrated on Fridays (and also on Tuesdays), while the priest heard confessions of penitents during that time. They also performed the Bitter Lamentations, held on Sunday afternoons during Lent. In smaller parishes, where a permanent organist was not employed, the congregation initiated the singing during the liturgy in all days of the year. The participation of singers in experiencing the Paschal Triduum was important. Traditional vigils (adorations) on the night of Maundy Thursday, the whole night from Good Friday to Holy Saturday (at the Holy Sepulchre) and on Easter night (almost until the end of the 20th century, the Blessed Sacrament was adored until Easter morning, i.e., until the resurrection procession) were conducted by leaders, who spent several hours a day in the church (Perszon 1992, pp. 24–34). The same people led the singing during the processions (Easter, blessing of fields on St. Mark's Day, Corpus Christi and its octave, to the cemetery on All Saints' Day and All Souls' Day).

The activity of the singers also had a wider, supra-parish dimension. This concerned participation in walking pilgrimages to local sanctuaries, which involved Kashubians on a larger scale in the past centuries. Neither parish documents nor family memories from the 19th century confirm organised Kashubian pilgrimages to Marian places; neither to Swarzewo (where a limited pilgrimage movement existed already in the 18th century) nor to Sianowo (here, regular walking parish pilgrimages were initiated in 1980, Pryczkowski 2016, pp. 259–65). For a long time, the only place of pilgrimage in Kashubia (since the mid-17th century) was Kalwaria Wejherowska (Kustusz 1991, p. 76). While the route and course of the Oliwa Pilgrimage (and earlier from Gdańsk–Stolzenberg) was included and recorded in the *Kalwaryjka*, i.e., the prayer book *Droga do nieba* (Eng. "Road to Heaven"), the equally old Kościerska Pilgrimage (Jażdżewski 2008) is not so clearly regulated. It can be assumed that its scenario (three days to Wejherowo, stay at the sanctuary and then three days back) was very similar to that of the Gdańsk (Oliwa) Pilgrimage. On the way traditional songs were sung, the Rosary was recited, and the "Angel of the Lord" (Pl. Anioł Pański) and litanies were sung (including the Litany of the Sweetest Name of Jesus). Hours of the Virgin Mary, Chaplet to Blessed Virgin Mary, and sermons (speeches) were always preached in the same places, and chapel and churches were visited along the way (**Droga do nieba, pp. 20–78**).

The pilgrimage companies were always accompanied by a brass band, which made the marching easier and lifted the spirits of the pilgrims. The march, however, was mostly filled with singing; it was led by singers walking in front of the group but behind the cross. Thanks to their loud voices, they engaged several hundred pilgrims in songs (casual, Marian and Passion pieces) and prayers. The same rules applied to shorter Kashubian pilgrimages: to Swarzewo, Wiele and Sianowo. In the 1980s, a new form of pilgrimage appeared in Kashubia (as well as throughout Poland): walking pilgrimages to Jasna Góra. They grew out of the Warsaw Walking Pilgrimage to Jasna Góra, in which priests and seminarians from Pomerania participated in a large number in the 1970s (Ryszka 2010). When, as a result of the election of Karol Wojtyła as the Pope (1978) and the "Solidarność Revolution" of 1980–1981, new pilgrimages from Pomerania were established (first in 1979, with the Pomeranian Pilgrimage from Toruń, then in 1982 from Swarzewo, in 1983 from Gdańsk, in 1986 from Gdynia, and finally, in 1991 from Kościerzyna), the style and organisation of the Warsaw Pilgrimage were adopted in Pomerania and Kashubia (Kamiński 2013). In the following years, all walking pilgrimages to local sanctuaries (Swarzewo, Wejherowo, Sianowo, Wiele) followed the patterns of the Jasna Góra pilgrimages.

Pilgrims coming to the Kalwaria fairs (most of them came to Wejherowo by means of transport: until the mid-20th century, by horse-drawn carts or by rail; from 1873, or if they were from the vicinity, they came on foot, in family or neighbourhood groups) participated in "celebrating stations", that is, the Passion service on Kalwaria. It includes a procession around Kalwaria (through 25 chapels) with meditation on the Passion of the Saviour. Between the stations, songs about the Lord's Passion and to the Mother of God are sung. For nearly 300 years, this singing was led by gathered singer–leaders. After the introduction of a portable sound system in the 1980s, and as a result of the gradual disappearance of "professional" singers, today, the entire service, lasting about 3 hours, is animated by the Franciscans, custodians of Kalwaria. The role of volunteer singers was, therefore, reduced.

An important function of the leader (formerly mainly men, with women today) was and still is to conduct May services, celebrated in the evening at village chapels and crosses. These traditions are still fostered in Kashubia, although the number of participants of this prayer is decreasing in many places. This may be related both to the depopulation of some hamlets and, above all, to the change in the social structure in many villages. Kashubia is an increasingly urbanised area (new housing estates of single-family houses), and the new, nominal inhabitants of the village almost never integrate with the local community. Hence, their "absence" in such services is not surprising. Thirty years ago, a significant decrease in interest in the Marian May service in the rural environment had already been observed

(Perszon 1993, pp. 35–52). A similar process applies to June services, still practiced in some places, dedicated to the Sacred Heart of Jesus. The tradition of neighbourly sung Sunday Vespers disappeared as well; it used to occur sporadically even in the 1950s. In more pious houses, however, Lamentations are still sung, especially when someone is sick at home or in villages far away from the church.

A very clear breakdown of the singing tradition concerns the "empty night". This is due to the social modernisation and "technologization" of everyday life, as well as the changing attitude to faith, and, consequently, to salvation. The medicalisation of life (of mentality) and dying (frequent death in a hospital or hospice), social "repression" of death, often coincides with the deprivation of its eschatological dimension. The progressing specialisation of social services has already definitely covered the celebration (rites) related to death and funeral. Professional services (hospital, funeral parlour and parish) effectively "excluded" the family and the neighbourhood community (if it still exists) from active, subjective participation in dying and funerals. Even where there are still local singers, ready to watch over the deceased selflessly, the family resigns from it (Perszon 2021, pp. 7–11). Thus, not only the home liturgy of "good death" is eliminated, but all its subsequent stages are as well: body preparation, Rosary services, empty night and walking procession to the church. Traditional singer–leaders are, thus, within their professional competence, redundant.

## 5. The Twilight of Traditional Religious Culture of Singing

In the 1970s, in the field of religiosity (and pious folk singing) in Poland (including Kashubia), dynamic changes were taking place. They were initiated much earlier, due to the modernisation of life and social changes after World War II (Ziółkowski 2002; Kocik 2002; Jawłowska 2002; Adamski 2002). In addition to the disappearance of the previously natural habit of singing at home, which had already been noticed by informants in the 1960s (the universality of the radio in homes, working outside the home, the phenomenon of peasant workers, youth commuting to schools in cities and the influence of the so-called controlled secularisation), in the following years, free time within families (especially in the evenings) was dominated by television. Its attractiveness (despite communist propaganda) was so great that it gradually "switched off" family relationships, made it difficult to pray together, and eliminated family singing. Folk culture (meetings, conversations, singing, village Sunday dances, making music together, village services, etc.) was quickly giving way to mass culture served by the media. Elderly people remained faithful to the old customs and habits, while the younger ones began to live in a new, different "world".

These phenomena, as a result of technological and social changes, became evident to sociologists as early as 30 years ago with the cracking of cultural (including religious) transmission in families, the disappearance of common home prayers, the distancing of younger people from traditional customs and rituals and the spiritual "desertification" of the younger generations (Piwowarski 2000, pp. 95–110, 265–99; Perszon 2021, p. 7). The revival of religious life among young people observed in Poland in the 1980s (due to mass oasis and pilgrimage movements, the popularity of John Paul II and the universal acceptance of the Catholic Church) is considered by sociologists to be the result of a symbiosis of religious and national elements (Mariański 2013, pp. 60–73; Mazurek and Potulski 2020; Grabowska 2018). The process of secularisation radically accelerated after the Internet was introduced to Poland. From the beginning of the 21st century, the process of erosion of cultural traditions (in the family, school, neighbourhood community and parish) had already become clear and, to date, it seems irreversible. The consequence is a growing lack of interest among young people in religious life; initially, it concerned the so-called optional practices, while later also extending to the basic ones (Sunday Mass, daily prayer) (Mariański 2006, pp. 185–96; Kaufmann 2004, pp. 111–31). New songs gradually started to enter churches and liturgy (e.g., Czarna Madonna, Barka, Była cicha i piękna). A separate genre of songs (especially liturgical) resulted in the Light-Life Movement (Czupryński 2011). Life of the old, traditional song depends on the invention of the organists, who have

had the "advantage" of the sound system since the 1980s. Considering the disappearance of religious singing at home, it is not surprising that younger Kashubians hardly know the old, traditional repertoire. Almost none of them use prayer books and songbooks anymore (Perszon 2015a, pp. 78–80).

The old pilgrimage songs have been replaced (with the exception of a few, for example, *Po górach, dolinach* (Eng. "Through the Mountains and Valleys")) with a new, rich and constantly growing repertoire of songs performed with guitar accompaniment. Pilgrims are equipped with songbooks, in which traditional songs are only on the margins (Bartkowski 1992, pp. 82–83). Therefore, over the past half century, a significant cultural change has taken place within the Catholic community.

Another radical shift was brought about by a two-year (2020–2022) epidemic of COVID-19. In 2021, social life, religious cult and family forms of piety were nearly completely eliminated. Common Rosaries for the deceased and empty nights were "suspended", and participations in the funeral were reduced to a few people; May and October services were not held at crosses and chapels; the singing of Hours of Virgin Mary before the Sunday High Mass and Bitter Lamentations in Lent were suspended as well. Both fairs in Kalwaria Wejherowska, as well as fairs in Swarzewo, Sianowo and Wiele, were cancelled, and so was the celebration of the Stations of the Cross. In the opinion of the informants, this "shock" contributed to a significant weakening of those forms of sung prayer that do not have an institutional basis. The empty nights are primarily meant here, which, according to the singers, have practically (and almost completely) disappeared in 2022.[36]

## 6. Kashubian Singing as a Performance

The weakening of the folk singing culture in Kashubia coincided with the artistic promotion of its most specific manifestation, i.e., the "empty night". It is this element that has become "fashionable" not only in the Pomeranian artistic world; however, not without a reason, it was considered exposed in ennobling such as in the celebration of the funeral of people valued in Kashubia. It happened at the turn of the 20th and 21st centuries. According to Eugeniusz Pryczkowski, at the end of the 20th century, people began to be interested in the vanishing ritual of the "empty night". The mystery "Dream about Europe" organised on 25th September 1998 in the church of St. Jan in Gdańsk contributed to this change. During the mystery, customs and rituals related to this Kashubian wake service were presented. Among the performers representing several nations, Kashubians presented themselves: singers from the Sianowo parish (Pryczkowski 2004a, p. 37). The event received extensive press and television coverage. The public artistic "promotion" of the Kashubian prayer for the dead aroused respect for the custom, which had sometimes been opposed by the clergy[37] in the past. Empty night has become a source of inspiration for writers, poets and amateur theatre groups[38]. A special circumstance promoting this rite was the death of John Paul II. Kashubian activists came up with the idea to organize an "empty night" vigil on the last night before the Pope's funeral. The choice fell on Sianowo, the central place of Kashubian religious worship and important for Kashubian regionalism. All the "singers" there are shaped in deep Marian devotion (Pryczkowski 2004b, pp. 147–52). The program included singing religious songs in the Kashubian language read and sung by professional artists. This vigil began with a solemn Mass for John Paul II with a liturgy in the Kashubian language and with the participation of the Chief authorities of the Polish Composers' Union. The Mass with the Kashubian Liturgy of the Word was celebrated by the priests Jan Perszon, Leszek Jażdżewski and Bogusław Głodowski, as well as the seminarians of the Theological Seminary in Gdańsk-Oliwa. The Mass was broadcast on public television throughout Poland.

In the breaks between the songs, on a special big screen, meetings of Kashubians with John Paul II were presented, especially his words addressed to Kashubians in Gdynia and Sopot. The mystery ended at midnight. This event was commented on by all regional media (Pryczkowski 2016, pp. 307–9). In May 2005, singers from the Sianowo parish were invited to the 3rd Concert on the occasion of the 15th anniversary of the television

programme "Rodnô Zemia" at the "Miniatura" theatre in Gdańsk. It was dedicated to the memory of John Paul II and Izabella Trojanowska, the first editor of the Kashubian broadcast. All four concerts gathered numerous fans of the programme and the Kashubian region (Pryczkowski 2016, p. 310). Additionally, a "public" empty night took place in Sianowo. It was preceded by the funeral of the custodian of the sanctuary, Fr. Waldemar Piepiórka, which took place on 15th February 2008. The singers from Sianowo played an active part in the funeral ceremonies, during which the deceased was posthumously awarded with the Star of the Order of Polonia Restituta by the President of the Republic of Poland, Lech Kaczyński. The singers from Sianowo also celebrated the "empty night" of the President of the Republic of Poland, Lech Kaczyński, on the day before his funeral. They prayed for all the victims of the plane crash on the 10th of April 2010 in Smolensk[39]. The same singers went to Banino on the 22nd of April this year to sing and pray in the local church for Admiral Andrzej Karweta—the commander of the Navy, who was buried a day later at the local cemetery. Many local parishioners sang along as well. (Pryczkowski 2016, p. 310).

On the 18th of March 2008, Television Gdańsk broadcast an extensive report from the empty night vigil in a real, original form. The recordings were carried out before the funeral of Mieczysław Pryczkowski from Staniszewo-Wybudowanie (Sianowo parish). The big family of the deceased gathered in the mourning house and about fifteen singers took part in the report and talked about the specificity of this rite known in the area[40].

Dozens of singers from Sianowo and the surrounding area took part in the empty night in Kartuzy, dedicated to the death of Wojciech Kiedrowski, an eminent Kashubian activist publisher, who died on the 18 April 2011. The topic was also discussed several times on the radio. In March 2012, a group of traditional music enthusiasts from the Trójmiasto organised the singing of empty nights songs at the Pilgrim's House in Sianowo for about twenty people, who lived or studied in Trójmiasto. Radio Gdańsk recorded the coverage from the meeting in Sianowo. Two years earlier, the same station broadcast a report about singers from Staniszewo. Singers from the parish of Sianowo took part in the production of the documentary film *Ze śmiercią na Ty* (Eng. With Death on a First-Name Basis) directed by Ewelina Karczewska. The film was screened at the Kashubian Film Festival in 2014 in Miszewko (Pryczkowski 2016, p. 311). On the 8 October, the day before the funeral of a famous medievalist, professor Gerard Labuda from Luzino, in the church of St. Lawrence in Luzino, the Holy Mass was held in his honour first, and then the local singers prayed by performing the "empty night"[41].

On the 20th of December 2013, at night, the parishioners from Sianowo and members of the Polish Composers' Association organised an "empty night" in the church of St. Jan in Gdańsk in the intention of the late professor Brunon Synak, an outstanding Kashubian activist, (Pryczkowski 2016, p. 310). The Museum of Kashubian–Pomeranian Literature and Music has been organising "Meetings with Kashubian music" for years. These events refer to the local traditions of religious singing, e.g., Empty Night or Kalwaria singing[42]. A sui generis cultural event was the spectacle "Empty Night. When you are leaving, soul", directly referring to Kashubian funeral rites. A group of singers from Luzino (from the Klein family) took an active part in the verbal and musical performance. Its premiere took place in the Collegiate Church of the Holy Trinity in Wejherowo on the 11th of October 2018[43]. The traditional repertoire (e.g., referring to the "empty night") appears in the choral music of Pomerania, for example, in the CD recordings of the "Lutnia" Choir from Luzino. In 2015, musicologist Dr. Sławomir Bronk from the Music Academy in Gdańsk recorded with the *Discantus* Chamber Choir from Gowidlino and the Schola Cantorum of the Music Academy in Gdańsk, album *Cierpiącym Duszom. Pieśni Pustej Nocy* (Eng. "To the Suffering Souls. The Songs of the Empty Night"). Another Kashubian musicologist and creator, Dr. Tomasz Fopke, is the author of libretto of the opera in the Kashubian Language, entitled *Rebecca* by Michał Dobrzyński (2014). He also composed *Pierszi Kaszëbsczi Pasja* (Eng. "The First Kashubian Passion") to the words of the Gospel according to Saint Mark

(2002), *Kashubian Mass for Choir and Devil's Violin* (2008), as well as texts and music for over 500 songs. The body of his versatile work has been recorded in over a hundred albums.

There are numerous artistic initiatives referring to Kashubian religious singing. Although they are not able to reactivate the spiritual culture of the past, they skilfully "transfer" its elements to contemporary culture and the high-end (artistic) culture, but they also inspire, at least regionally, the popular (mass) culture. This proves the vitality of culture, whose representatives skilfully bring its heritage into the reality of the 21st century.

## 7. Conclusions

The above considerations allow us to conclude that religious folk singing has existed in Kashubia for at least five hundred years. Its fate is closely intertwined with the history of Poland and the Catholic Church in Pomerania. It developed parallel to the emergence and flourishing of literary Polish language, religious poetry and in dialogue with the Catholic Latin liturgy and Gregorian Chant. It matured and strengthened in the face of a dispute with Protestantism. Religious singing among Kashubians was also affected by political changes, a century and a half of Prussian partition and, finally, by 45 years of communist oppression. While songbooks were published periodically during the Reign of Prussia, which literally crossed the thresholds of people's houses and promoted religious singing in family life and local communities, the times of the Polish People's Republic brutally put an end to this process. However, the crisis of spiritual/religious singing came not only from the lack of popular hymnals; rather, the reasons for it are of a social nature (economic, technological, social, intellectual and moral changes; breakdown of cultural traditions in families) or they result from changes in the Church itself: the abandonment of the traditional repertoire in favour of the so-called new hymns and religious songs; changing the model of walking pilgrimages; coming to terms with secularisation trends; the lack of one adequate recipe for secularisation and the process of religious differentiation in society. The disappearance of the singing tradition must also be associated with the culture of free time. Mass culture prefers not to follow the beaten track of tradition, but to "look for" its own path, personal spirituality, constructing life according to the proposed (especially by electronic media) patterns (Steensland et al. 2018; Wilkins-Laflamme 2021; Smith and Cragun 2019; Starr et al. 2019; Levin 2022). Therefore, the "old", even if valuable, deep and beautiful, is no longer assimilated automatically (through habit) but requires making a decision to do so. Against this background, artistic initiatives (mysteries, concerts, events) that draw heavily on the singing tradition, giving it a new, attractive face, should be considered valuable and compelling.

The considerations do not address the otherwise very interesting issue of the development of religious songs in the Kashubian language, which is becoming a "new tradition" for many ethnically awoken communities.

**Funding:** This research received no external funding.

**Institutional Review Board Statement:** Not applicable.

**Informed Consent Statement:** Not applicable.

**Conflicts of Interest:** The author declares no conflict of interest.

## Notes

[1]    The musicologist (Zoła 2010, p. 372) notices that, until recently, in the folk way of celebrating a ritual (liturgical) year, time was measured not by means of dates but a holiday that fell on a particular day.

[2]    The Synod of Włocławek (Kashubia belonged to the Pomeranian Archidiaconate in the Diocese of Włocławek) of 1641 admonishes the congregation to sing *Dziesięcioro rytmem polskim* (*Ten Commandments to the Polish Rhythm*) on Sundays after the sermon.

[3]    In the Poor Clares nunnery in Stary Sącz, Kinga of Poland, as indicated in her hagiography (she died in 1292), before leaving the church, sang 10 chants "in vulgari", that is, the national language.

[4]    The Synod hosted, inter alia, the Bishops of Wrocław, Poznań, Cracow, Włocławek, Płock, Lubusz Land, Chełmno. In 1285, at the Synod of Łęczyca, Archbishop Jakub Świnka ordered the priests to say (but also sing) and explicate the Apostolic Creed, Lord's

Prayer and Hail Mary in Polish. As early as the 14th century, singing of the Decalogue and a prayer was an integral part of the Eucharistic liturgy.

5    Reminding people about the Hail Mary prayer was a preachers' duty, which is confirmed by preserved Dominican and Franciscan texts from the 16th century.

6    For a long time, ritual songs were chanted in Latin. The clergy in Poland during the Piast dynasty consisted of people of Czech, Bavarian, Saxon, Italian and Rhenish origin. Therefore, the knowledge of the Slavic language was a prerequisite for Christianisation. It was already required from Five Martyr Brothers and Bruno of Querfurt (1001–1003).

7    Songs in the national language were sung instead of the existing Latin psalms together with the people and after the sermon.

8    Disasters afflicting European societies (Hundred Years' War, plagues, famine, Great Schism) escalated the atmosphere of horror and the fascination with Dolorism. Since the 13th century, one of the elements of such an atmosphere in Poland was the movement of flagellators, the cult of the Five Holy Wounds, Our Lady of Sorrows, liturgical dramas and religious mystery plays. The Church ultimately forbade such spectacles after the Council of Trent.

9    They included songs intended for subsequent periods of a liturgical year, processions and Eucharistic devotions, vespers, little hours and litanies.

10   At the turn of the 17th and 18th centuries, the chanted catechism (written in verse) was an integral part of the Sunday Catholic liturgy. It was all sung before Holy Mass; when the congregation did not manage to sing all the texts, the singing was finished after the Mass. During the Mass itself, singing also took place. After the sermon, the congregation sang the Apostles' Creed, another song between elevation of the host and the Lord's Prayer, and another one after the elevation. See (Bielawska 1980, pp. 128–31). In Kashubia, at the turn of the 16th and 17th centuries, guarding the Polish singing tradition was a responsibility of the Bishops of Włocławek (Hieronim Rozrażewski, the Synod of Włocławek of 1641).

11   The liturgist from Lublin informs that the chants complemented or replaced the tradition of singing the *Lord's Prayer, Hail Mary*, the *Apostles' Creed* and the *Decalogue* in temples.

12   Little Hours are the equivalent of Latin *horae*, that is a breviary consisting of antiphonies, hymn, psalms and Bible readings. In the Middle Ages, small offices (*officium parvum*), that is, shortened breviaries for laymen (tertiaries and members of brotherhoods), were composed. Hours of the Passion Polish texts come from the 15th century. Versed biblical stories and songs of the Saints were known in the Middle Ages too. See (Korolko 1980, pp. XLIV–L).

13   The Little Office of Immaculate Conception published in 1616 is a translation of the Spanish text written by a Jesuit priest Alfonso Rodriguez. In 1615 the "Polish" Little Office was approved by the Holy See.

14   Wit, 264–265. The Passion devotion chanted in Polish developed in the 18th century in the Holy Cross Church in Warsaw (Brotherhood of St. Roch), from where it spread over the entire country. Compare (Bańbuła and Bartkowski 1989, pp. 1309–311).

15   The author emphasises that the development of Polish Catholic pious singing was to a great degree triggered by the development of Protestantism, which reached people by means of Polish editions of psalters, hymnals, prayer books and the Bible. In order to read about the diversity of Protestant pious singing in Poland, see (Nowicka-Jeżowa 1992, pp. 29–237).

16   One of the forms of this devotion was practised by scapular brotherhoods. Rosaries, usually made with a cord and beads, helped in praying chaplets. In order to read about Carmelite brotherhoods wearing the scapular and spreading the idea of serving Virgin Mary, see (Kopeć 1997, pp. 362–66).

17   Kopeć adds that the victory of the Christian fleet over Turks in the Battle of Lepanto (1571) and establishing the celebration of Our Lady of Victories by Pope Pius V were a strong impulse in propagating the Rosary. Pope Gregory XIII changed the day of the celebration for the first Sunday of October (Our Lady of the Rosary), whereas in 1577, the Master of the Order of Preachers gained the right to found Rosary confraternities in parishes. It very quickly resulted in the popularisation of brotherhoods and the Rosary prayer among the people.

18   The songbook started to be composed by Walentyn Goltman, a student from Wejherowo, and was finished by Father Franciszek Ruthen, a Wejherowo parish priest.

19   The former one includes 1218 songs. Because of a high demand among the congregation, it had a few reeditions. The reprint of the second edition from 1886 came out in Pelplin (*Bernardinum* Publishing House) in 2015.

20   After the monastery in Wejherowo was dissoluted by Prussians in the middle of the 19th century, the prayer book was renewed by the curia of Chełmno in Pelplin. See (*Droga do nieba czyli Rozpamiętywanie Męki P. Jezusowej i innych tajemnic św. w pobożnej pielgrzymce po Kalwaryi będącej pod zarządem OO. Reformatów przy mieście Wejherowie z przydatkiem niektórych pieśni o modlitw nabożnych* 1885), E. Michałowski Bookstores, Pelplin, 1885.

21   Among songbooks and prayer books published in Pelplin at the turn of the 19th and 20th centuries, Father Edward Hinz (257) enumerates as many as 19 of them, i.a., a book by Father B. Ruchniewicz and the renewal of the Wejherowo calvary *Droga do nieba* [Eng. "Road to Heaven"] from 1909. It proves intensified piety in the Diocese of Chełmno and intense singing activity of Catholic laity.

22   The edition of a handy songbook was handled by a local Kashubian landowner, Jan Herman Szczypior, who wanted to make it easier for his singing companions to pray by the dead people. See Nowy kancjonał pieśni nabożnych wedle obrządku Kościoła katolickiego na uroczystości całego roku. Poznań 1908.

23    Labuda (109) claims that in the 17th and 18th centuries, the Pole–Catholic and German–Protestant stereotypes are inadequate since it was Kashubians who were Catholics and Pomeranians who were Protestants then, regardless of their nationality. It was after Gdańsk Pomerania was taken over by Prussians that Lutherans started to tilt to the Prussian state, whereas Catholics started to favour the "Polish" Church.

24    The essence of the medieval and posterior religious poetry was the so-called versus rhytmicus, which allowed easy memorisation. Such rhythmicity (kind of a rhyme, syllabic verse) was also commonly applied in religious chants and recitations in Polish (Korolko 1980, pp. LIX–LXIV).

25    Similar conclusions are drawn in the monograph devoted to polyphonic music in Lower Silesia by Pośpiech (2004, pp. 66–72).

26    Parish (municipal) schools in the Middle Ages educated their students in music as one of the seven liberal arts. See: (Hinz 1994, pp. 49–53). The musicologist of Pelplin describes the history of church education in Chełmno by the Vistula.

27    The author cites the history of this grassroots movement, initiated in Germany around 1830, aimed at renewing Church music and restoring its liturgical character. In Poland, the Cecilian movement took the name *Towarzystwo św. Wojciecha ku Podniesieniu Muzyki i Śpiewu Kościelnego* (Eng. Society of St. Wojciech for the Rebirth of Music and Church Singing)

28    Genealogy of the Klein family from Barłomino (Sampowskie). Typescript in author's possession.

29    The Church choir in Luzino included: Jadwiga, Maria and Gertruda Klein (before 1939) and after the war: Leon, Jerzy, Regina Klein. Currently, the members of the Lutnia Choir are: Wojciech, Stanisław, Krystyna Klein, Magdalena Gajowska (nee Klein). Inf. Roman Klein, Barłomino-Sampowskie. 22nd September 2002.

30    In addition to the liturgical calendar, folk singing in Poland was regulated by the days of the week. Monday and Tuesday were devoted to the worship of angels and Saints, Wednesday to St. Joseph, Thursday to the Eucharist, Friday to the Passion of the Lord, Saturday to the Blessed Virgin Mary and Sunday to the Holy Trinity. See: (Zoła 2010, p. 373).

31    Inf. Teresa Trybowska, nee Klein from Luzino. 22nd September 2022.

32    A monograph on the Miotk family was written by Andrzej Miotk (2009).

33    Inf. Eugeniusz Pryczkowski from Banino. 30 September 2022.

34    See note 33 above.

35    Rosary is the most common home prayer (of family and personal kind). In many churches, lay people pray the Rosary before the everyday morning or evening service; Rosary (sang or recited) is a prayer said during a pilgrimage on foot (or by coach) to a shrine; the Rosary is the basis of the prayer for the deceased (funeral); it is also the core of prayers during days of recollection and closed retreats.

36    Inf. Stanisław Miotk from Sianowo-Kępa; Fr. Marian Miotk from Gniewino; Jan Pryczkowski and Edmund Jóskowski from Stążki; Stanisław and Wojciech Klein from Luzino; Roman and Bernard Klein from Barłomino; Brunon and Łucjan Krefta from Gowidlino; Marian Szymański from Grabowiec; Jadwiga Serkowska and Kazimierz Lubecki from Lipusz; Konrad Jank from Klukowa Huta; Jan Groth from Darzlubie; Józef, Łukasz, Marieta and Teresa Koszałka from Załakowo.

37    Rodnô Zemia of 18th March 2008. In this program, Fr. Stanisław Majkowski, the custodian of the sanctuary in Swarzewo, said that before the war, the ritual of the empty night was stigmatised by the contemporary parish priest, Fr. Wojciech Pronobis, as a result of which it disappeared from Kępa Swarzewska;

38    For example, the "Dialogus" Theatre from Parchowo under the direction of Jaromir Szroeder, prepared a spectacle based on the rite of the empty night. See: Rodnô Zemia, 17 April 2005.

39    J. Dosz, *Region Kaszuby w obliczu wielkiej narodowej tragedii*, "Norda", 23 April 2010.

40    Rodnô Zemia of 18th March 2008. Inf. Eugeniusz Pryczkowski from Banino.

41    Inf. Józef Borzyszkowski from Gdańsk; Jerzy Klein from Luzino.

42    Inf. Witosława Frankowska from Wejherowo; Tomasz Fopke from Wejherowo.

43    Inf. Witosława Frankowska from Wejherowo; Roman Klein from Barłomino; Wojciech Klein and Władysław Kankowski from Luzino. The spectacle was directed by Wojciech Rybakowski, the author of the text is Jerzy Stachurski, and the author of the music is Cezary Paciorek.

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
