# Peer review of "Religious Singing in Kashubia: Tradition and Modernity"

_religions, doi:10.3390/rel14020231_

Round 1

Reviewer 1 Report

The writing style that includes many clauses in a single sentence makes it hard to read. I would recommend one idea one sentence approach and avoid packing in loads of information in one sentence. It makes the essay unduly complex. Also, the use of certain local terms e.g. temple, and the May event are not clear to learned general readership. Christians tend not to commonly designate their sacred space as "temple." It is also unclear what is the May event. So, some explanation perhaps in footnote might be helpful. Also, lay Catholics are not known to do much singing and often relegate this activity to parish choirs and cantors. To that end, I think blurring the line between official liturgical services of music-making, and folk liturgical musical practices further complicates this investigative historical essay. Perhaps a clearer roadmap that just focus on folk liturgical music-making might be better. Perhaps the journal editor can assist in refining this essay - making it simpler to read and helping to minimize its textual complexity.

Author Response

Thank you for your specific comments. According to the feedback, the text has been corrected by a native speaker, i.e. compound sentences have been replaced by singular ones. As far as terminology is concerned: the division of church songs into liturgical and non-liturgical, as well as into religious but privately performed, is clear and transparent in Polish musicology and ethnomusicology. Throughout Poland (especially in the countryside), the lay faithful know and sing numerous religious songs, without distinguishing between 'liturgical' and 'extra-liturgical' songs. Religious musical folklore is thus an extension of the ecclesiastical repertoire.

Reviewer 2 Report

It is a very interesting article with a great potential that has not been revealed in the sufficient level. Firstly, there is a lack of methodological paragraph. Author(s) has/have not explained methods used in the research, has/have not formulated the hypothesis and research questions. Secondly, there should appear some information about Kashubia and Kasubian culture. In these form this paper is addressed only to Polish readers or narrow group of specialists in the Kasubian or Polish studies. Author(s) should have referred to literature dedicated to folk religion in Poland and Kashubia, e.g. to the brand new work of Monika Mazurek and Jakub Potulski. Finally, Author(s) should more strongly link findings of his/her/their research with literature in the Conclusions.

Author Response

Thank you very much for your comments, which will make the text professional. In the study of the so-called folk piety in Kashubia, I cannot very much refer to other studies, as no one else is currently undertaking them to a greater extent. I have included some notes on methodology and suggested literature in the text. Widening the spectrum of comparisons to include other regions in Poland would lead to a considerable increase in the size of the article.